# Nature-Based Early Childhood Education and Children’s Social, Emotional and Cognitive Development: A Mixed-Methods Systematic Review

**DOI:** 10.3390/ijerph19105967

**Published:** 2022-05-13

**Authors:** Avril Johnstone, Anne Martin, Rita Cordovil, Ingunn Fjørtoft, Susanna Iivonen, Boris Jidovtseff, Frederico Lopes, John J. Reilly, Hilary Thomson, Valerie Wells, Paul McCrorie

**Affiliations:** 1MRC/CSO Social and Public Health Sciences Unit, University of Glasgow, Berkeley Square, 99 Berkeley Street, Glasgow G3 7HR, UK; avril.johnstone@glasgow.ac.uk (A.J.); anne.martin@glasgow.ac.uk (A.M.); hilary.thomson@glasgow.ac.uk (H.T.); valerie.wells@glasgow.ac.uk (V.W.); 2CIPER, Faculdade de Motricidade Humana, Universidade de Lisboa, Estrada da Costa, Cruz Quebrada, 1499-002 Lisboa, Portugal; ritacordovil@fmh.ulisboa.pt; 3Faculty of Humanities, Sports and Educational Sciences, University of South-Eastern Norway, 3672 Notodden, Norway; ingunn.fjortoft@usn.no; 4School of Applied Educational Science and Teacher Education, University of Eastern Finland, 80101 Joensuu, Finland; susanna.iivonen@uef.fi; 5Research Unit for a Life-Course Perspective on Health and Education, Department of Sport and Rehabilitation Sciences, University of Liege, 2 Allee des Sports, 4000 Liege, Belgium; b.jidovtseff@uliege.be; 6Laboratory of Motor Behavior, Faculdade de Motricidade Humana, Universidade de Lisboa, Estrada da Costa, Cruz Quebrada, 1499-002 Lisbon, Portugal; fred.lopes3@gmail.com; 7School of Psychological Sciences and Health, University of Strathclyde, 50 George Street, Glasgow G1 1QE, UK; john.j.reilly@strath.ac.uk

**Keywords:** early childhood education, children, preschool, social, emotional, cognitive, nature

## Abstract

This systematic review synthesised evidence on associations between nature-based early childhood education (ECE) and children’s social, emotional, and cognitive development. A search of nine databases was concluded in August 2020. Studies were eligible if: (a) children (2–7 years) attended ECE, (b) ECE integrated nature, and (c) assessed child-level outcomes. Two reviewers independently screened full-text articles and assessed study quality. Synthesis included effect direction, thematic analysis, and results-based convergent synthesis. One thousand three hundred and seventy full-text articles were screened, and 36 (26 quantitative; 9 qualitative; 1 mixed-methods) studies were eligible. Quantitative outcomes were cognitive (*n* = 11), social and emotional (*n* = 13), nature connectedness (*n* = 9), and play (*n* = 10). Studies included controlled (*n* = 6)/uncontrolled (*n* = 6) before-after, and cross-sectional (*n* = 15) designs. Based on very low certainty of the evidence, there were positive associations between nature-based ECE and self-regulation, social skills, social and emotional development, nature relatedness, awareness of nature, and play interaction. Inconsistent associations were found for attention, attachment, initiative, environmentally responsible behaviour, and play disruption/disconnection. Qualitative studies (*n* = 10) noted that nature-based ECE afforded opportunities for play, socialising, and creativity. Nature-based ECE may improve some childhood development outcomes, however, high-quality experimental designs describing the dose and quality of nature are needed to explore the hypothesised pathways connecting nature-based ECE to childhood development (Systematic Review Registration**:** CRD42019152582).

## 1. Introduction

The foundations of good cognitive, social and emotional health are established from an early age [1]. However, according to a global report by the World Health Organization (WHO), evidence suggests that cognitive, social, and emotional outcomes are often low in children, and typically worse in those with lower socio-economic status (SES) [2]. Children with social and emotional problems tend to be more likely to have poorer physical health, relationship problems, lower academic achievement in school, and continued mental health problems as they mature into adulthood [3,4,5]. Therefore, it is important to intervene early in the child’s life when they are rapidly developing, with universal early childhood education (ECE) being particularly important for healthy childhood development [1,6].

Traditional ECE settings are typically characterised by children spending most of their time indoors and when they are outdoors the playground tends to consist of predominately man-made structures, such as swings, climbing frames, and slides with limited integration of natural elements, such as trees and woodland areas, varied topography and vegetation with a variety of trees, rocks, fallen logs, and loose parts [7,8]. In contrast, in nature-based ECE (also known as forest pre-schools, forest kindergartens, nature-based pre-schools, etc.) nature is integrated into the philosophy, curriculum and/or wider environment, and children typically spend the majority of their day outdoors in immersive nature experiences [9]. These types of settings, with their rich, diverse, and outdoor promoting environment, may provide additional benefits to children’s health, wellbeing, and development beyond the traditional ECE provision [9,10]. 

Evidence suggests that exposure to nature in its broadest sense (i.e., community, home, and educational environments) may be particularly beneficial in improving children’s, social, emotional, and cognitive development [11,12,13]. Two systematic reviews exploring the effect of nature on children and adolescents’ health suggested improvements in emotional wellbeing, overall mental health, resilience, self-esteem, and reduced stress [11,12,13]. Nature-based ECE settings may provide additional affordances that are more dynamic and changeable in comparison to traditional ECE where affordances are more fixed. Affordances relate to the opportunities provided by the environment and how they are perceived and interacted with by an individual according to their individual capabilities [14,15,16,17]. For some children, a tree can provide a climbing opportunity, and for others it may provide shelter or be included in different games during their play. Importantly, these affordances may be influenced by changing seasons, weather, and a moving environment. Providing more varied and dynamic affordances in nature-based ECE compared to traditional ECE may facilitate children engaging in a broader range of play types (active, risky, imaginative, etc.), and these are integral to social and emotional development. When children play in nature, they socialise with their peers more and engage in higher levels of physical activity [18], which in turn, may impact, social, emotional, and cognitive outcomes.

Despite the potential of nature-based ECE for improving children’s social, emotional, and cognitive development, the evidence base is currently limited. Presently, much of the existing systematic review literature focuses on a wider age group (2–18 years), the impact of exposure to nature in the broadest sense (i.e., not solely nature-based ECE), and physical outcomes. Additionally, where evidence does exist, a high risk of bias is present [11,12,13]. Conducting a systematic review of nature-based ECE on children’s health and development, specifically, will inform future research needs, synthesise and summarise the global evidence, inform national and international policy, and potentially enable the field to understand the mechanisms by which certain child-health outcomes might improve. 

To our knowledge, this will be the first systematic synthesis of **nature-based ECE** on young children’s (2–7 years) **social, emotional, and cognitive** development. By synthesising both quantitative and qualitative evidence, this review can provide a comprehensive understanding of the evidence base that could be pivotal to informing future research, policy, and practice in the nature-based ECE field. Therefore, the aim of this mixed-methods review was to:(a)Determine if attending nature-based ECE is associated with children’s social, emotional, and cognitive development.(b)Explore children’s, parent’s and/or practitioner’s perceptions of nature-based ECE on children’s social, emotional, and cognitive development.

This systematic review used a novel mixed methods approach. The qualitative studies enabled a better understanding of the phenomenon of nature-based ECE and the quantitative studies were used to understand the impact on children’s development. This mixed method approach combines the strengths and limitations of research enquires [19].

## 2. Materials and Methods

This systematic review is part of a larger research project synthesising evidence on the association between nature-based ECE and children’s overall health and development [19]. Findings for other outcomes (i.e., physical health outcomes) have been published in the Journal of Physical Activity and Health. This systematic review was registered to the International Prospective Register of Systematic Reviews (CRD42019152582) in October 2019, and the protocol was published to BMC Systematic Reviews in September 2020 [10]. It follows the reporting guidance provided in the Adapted PRISMA for reporting systematic reviews of qualitative and quantitative evidence [20].

### 2.1. Eligibility Criteria 

The selection criteria followed the PI(E)COS (**P**opulation, **I**ntervention or **E**xposure, **C**omparison, **O**utcomes and **S**tudy design) framework [21].

*Population:* Children aged 2–7 years attending ECE and who have not started primary or elementary school education were included. The typical age of children attending ECE, accounting for global differences, is aged 2–7 years. To assess eligibility, the mean age, range, or median reported in the study was used. Studies that only included children with a disease or condition, such as autism, a physical disability, attention deficit hyperactivity disorder, autism, etc., were excluded. 

*Exposure/Intervention:* To be eligible, ECE settings, such as nature-preschools, forest kindergartens, forest schools, etc. [9], had to integrate nature into their environment. This may have included a spectrum of nature exposure, including: (i) children spending most of the ECE day outdoors in highly naturalised areas; (ii) interventions that enhance the amount and diversity of nature in the playgrounds; (iii) associations of specific natural elements (e.g., hills, trees, grass, vegetation, etc.) in the ECE setting; and (iv) the introduction of a garden-based exposure. Studies were excluded if the exposure was traditional ECE where children typically spend more time indoors and their environment was predominately manmade structures such as slides, swings, or climbing frames. 

*Comparison:* Traditional ECE was identified if children typically spent more time indoors and had fewer opportunities in environments that were not predominantly nature-based. These outdoor areas tended to integrate a small amount of nature with little variety, and incorporated manmade structures, such as swings, slides, and climbing frames. 

*Outcomes:* Any child-level outcome related to children’s social, emotional and cognitive health, wellbeing, and development were included. Studies were excluded if they assessed non-child level outcomes, such as the impact on practitioners, changes (i.e., outcomes) to the ECE settings, and studies using unvalidated questionnaires (for both quantitative and qualitative designs). 

*Study designs:* Quantitative and qualitative primary research designs were included. For qualitative studies to be eligible, they had to explore parent, practitioner, and/or child perceptions on children’s social, emotional and cognitive health, wellbeing, and development when the child attended nature-based ECE. For quantitative studies to be eligible, outcomes had to be measured when children attended nature-based ECE. For example, cross-sectional and case-control studies that measured outcomes when children were attending nature-based ECE; longitudinal, quasi-experimental, and experimental studies with at least two time points (before and after), and retrospective studies if outcomes were assessed when the child was attending nature-based ECE. Qualitative studies were excluded if they did not have: a comparator (i.e., exposure, control group, pre/post), addressed questions that were suitable for qualitative enquiry, and/or made a useful contribution to this review (see **Quality appraisal of included studies)**. 

### 2.2. Information Sources and Search Strategy

In October 2019, nine relevant electronic databases were searched (from inception onwards): 1. Education Research Information Centre (ERIC)—(EBSCOhost); 2. Australian Education Index—(Proquest); 3. British Education Index—(EBSCOhost); 4. Child Development and Adolescent Studies—(EBSCOhost); 5. Applied Social Sciences Index and Abstracts—(Proquest); 6. PsycINFO—(EBSCOhost); 7. MEDLINE—(EBSCOhost); 8. SportDiscus—(EBSCOhost); and 9. Scopus (Elsevier). 

Dissertation and Theses Database (ProQuest), Open Grey (www.opengrey.eu, accessed on 8 December 2019) were searched for grey literature, and Directory of Open Access Journals (www.doaj.org, accessed on 8 December 2019) to capture dissertations and reports. The first 10 pages of Google Scholar were searched and checked, and websites of related organisations and professional bodies involved in nature-based ECE were searched for relevant publications. Finally, in August 2020, citation lists of eligible studies published from 2019 onwards were screened to capture published evidence that may have been missed in the initial searches.

Two authors (AJ and AM) and an information scientist (VW) constructed the search strategies. A comprehensive search strategy was developed by reviewing keywords and related terms in relevant systematic reviews and publications. Co-authors who have expertise in fields related to nature, child health, wellbeing and development, education, and systematic review methodology reviewed and refined the draft searches. The strategy was tested and refined until a finalised search strategy was developed. Search strategies were adapted for each database and other web searches. The literature search was not restricted by year of publication or language. A draft search strategy for MEDLINE can be accessed in the published protocol [10] or Appendix A. References were imported to Endnote and duplicates were removed by one reviewer (AJ). 

### 2.3. Selection Procedure

Each title and abstract were screened by one reviewer (AJ, PM, RC, IF, SI, FL, BJ, VW) with a random 10% screened in duplicate independently (AM) to reduce bias. Using Covidence (www.covidence.org/, accessed on 14 January 2020) software, full-text articles were screened by two researchers independently. A third reviewer resolved any disagreement (AM) in instances when reviewers disagreed. Where multiple publications were reported for the same study, they were combined and reported as a single study.

### 2.4. Data Extraction 

Data from eligible studies were extracted by one reviewer (AJ) and cross-checked by another reviewer (AM, PM or HT). 

For **quantitative studies**, the following information was extracted:Study ID (authors, year of publication)Country of originStudy design.)Participants (age, gender, socioeconomic status, sample size, etc.)Intervention/exposure type and duration and information on what the comparator groups received.Outcome measures (type, assessment tool, time point of assessment, etc.)Outcomes and results (effect estimates, standard deviation, confidence intervals, etc.)

For **qualitative studies**, the following information was extracted:
Study ID (authors, year of publication)Country of originParticipants (as above)Intervention/exposure type and durationResearch aimsOutcome measures (interviews, focus groups, etc.)Outcomes and results (summary of key themes derived from data extractor and author).

Primary study authors were not contacted to obtain missing information due to constraints on time and the large volume of studies.

### 2.5. Quality Appraisal of Included Studies

The quality of all included studies was assessed at study level by two reviewers independently (AJ, AM, PM, HT), and disagreements were resolved through discussion. The Effective Public Health Practice Project (EPHPP) Quality Assessment Tool [22] was used to assess the quality of **quantitative** studies. The EPHPP tool is a commonly used quality appraisal tool in public health that assesses quality across a variety of quantitative study designs [22]. Minor modifications were made to the tool to ensure its relevancy for the present review, for example, defining the target population, specifying confounders of interest, and enhancing the overall rating of the paper (see Appendix A). 

The Dixon-Woods (2004) checklist [23] was used to assess the trustworthiness of eligible **qualitative** studies, which provides a set of prompts that were designed to appraise aspects of qualitative methodology. Studies were excluded from the review if the research questions were assessed to be unsuited to qualitative inquiry (question 2) or if the paper was assessed as not making a useful contribution to the review question (question 7) (see Appendix A). 

### 2.6. Data Synthesis

Data synthesis was done in three stages. Firstly, for quantitative studies a meta-analysis and sensitivity analysis (where studies with a high risk of bias were removed) was originally planned, however, calculating an overall effect size estimate could not be performed because only a small number of studies could be combined, studies were heterogenous (as interpreted by the I^2^ statistic) and/or studies did not provide the appropriate data to support standardising.

As a meta-analysis could not be conducted, a Synthesis Without Meta-analysis (SWiM) based on effect direction was performed [24]. The effect direction plot was used to summarise findings at both a **study level** and an **outcome level**. Study level effect direction plots are presented in instances where an outcome is reported in two or more studies with the same exposure category (explained below). Study level effect directions are then synthesised to provide a summary effect direction at an outcome level where study quality (as rated using the EPHPP tool), design, and sample size of studies are considered. For example, if three studies were synthesised for the summary effect direction plot and one study was rated as moderate quality (according to the EPHPP tool), was of a controlled before and after design, and/or had a larger sample size, then this study would hold a greater weighting compared to studies with lower quality, poorer study designs, and small sample sizes. The synthesis by effect direction addresses the question of whether there is evidence of a positive or negative association. In addition, a narrative summarising the effect direction at a study level was conducted where an outcome could not be grouped in the effect direction plot, i.e., because the outcome was only measured in one study within the same exposure category.

Outcomes were grouped into three domains after study selection: (i) social and emotional development; (ii) cognitive development; and (iii) nature connectedness. Social and emotional development presents two sub-domains: social and emotional, and play. Exposure categories were also generated after study selection based on exposure descriptions by study authors in the eligible studies. These were: (i) nature-based ECE, (ii) ECE natural playgrounds, (iii) natural elements within ECE, and (iv) garden-based interventions. Table 1 provides an overview of these exposure categories. 

Sub-group analyses to investigate differential associations (by age, gender, duration spent in ECE, etc.) were initially planned, however, the eligible studies did not provide sufficient detail to enable us to conduct sub-group analyses.

Secondly, for qualitative studies, we conducted a thematic analysis of author reported conclusions and participant quotes grouping data into higher and lower order themes. One reviewer (AJ) analysed the data inductively, generating themes that were discussed with another two reviewers (AM, PM) who checked themes and clustering against quotes (both authors’ conclusion and participant quotes, were reported). 

Finally, using a conceptual matrix [19], we integrated the syntheses of both qualitative and quantitative studies. Findings from the synthesis of quantitative studies were mapped against the themes from qualitative studies identifying confirmative and contradicting findings. Findings from the qualitative synthesis were also used to hypothesise potential mechanisms of why or how quantitative results might have occurred.

### 2.7. Certainty of Quantitative Evidence 

To assess the certainty of the evidence across studies at an outcome level, the Grading of Recommendations Assessment, Development and Evaluation (GRADE) framework was used [25]. The risk of bias, precision, consistency, and directness was assessed where two or more studies reporting on the same outcome were grouped. Based on these assessments, the certainty of evidence was upgraded or downgraded to provide an overall rating of very low, low, moderate, or high [25] for each outcome. Given the absence of randomised controlled trials (RCTs), the start rating was always low, however, as per GRADE guidance, ratings could be upgraded. Publication bias could not be assessed as there were not enough studies grouped per outcome. 

## 3. Results

### 3.1. Results of the Literature Search

Figure 1 presents the summary of results from the systematic literature search. After duplicates were removed, 31,098 articles remained, of which 29,729 irrelevant titles and abstracts were excluded leaving 1370 full-text articles to be screened. 1224 full-text articles were excluded for reasons detailed in Figure 1. Seventy qualitative studies were removed because they did not have a comparator (i.e., exposure, control group, pre/post) as did a further 11 studies after having their trustworthiness assessed. A total of 59 unique studies (representing 65 individual papers) met the inclusion criteria (including the physical outcome domain which is presented in another paper), of which, 36 studies (representing 40 individual papers) included a **social and emotional, cognitive, and/or nature connectedness outcome** (26 quantitative; 9 qualitative; 1 mixed-methods). 

### 3.2. Characteristics of the Eligible Studies

Characteristics of included studies are described Appendix A.

#### 3.2.1. Geographical Location

The majority of the studies were conducted in the USA (*n* = 11) [26,27,28,29,30,31,32,33,34,35,36,37,38,39,40], Australia (*n* = 5) [41,42,43,44,45] and Canada (*n* = 4) [46,47,48,49]. Three studies each were conducted in Norway [50,51,52], Sweden [53,54,55], and the United Kingdom [56,57,58]. Two studies were conducted in Italy [59,60] and the remaining studies were conducted in Finland [61], Germany [62], Poland [63], South Korea [64], and Turkey [65] (*n* = 1 study per country). 

#### 3.2.2. Study Designs

Of the quantitative studies (*n* = 27, including 1 mixed method study), most study designs were cross-sectional (*n* = 8) and controlled cross-sectional (*n* = 7) and there were fewer uncontrolled before and after (*n* = 6) and controlled before and after (*n* = 6). Ten studies were included in the qualitative thematic analysis (including 1 mixed method study). Of the qualitative studies, the majority of studies (*n* = 4) used observational methods only [40,44,45,51] or (*n* = 2) observation and interviews (structured or semi-structured) [39,52]. One study used interviews and teacher case studies [58], *n* = 1 used a combination of focus groups, interviews, observation, and artefact collection [49], *n* = 1 used a combination of interviews and surveys [61] and *n* = 1 used a combination of photo preference, drawings from children and interviews [37].

#### 3.2.3. Exposures

The preliminary synthesis of eligible studies indicated that the included quantitative and qualitative studies could be grouped into four different exposure categories (as described in Table 1): nature-based ECE (*n* = 20), ECE natural playgrounds (*n* = 10), natural elements within ECE (*n* = 4) and garden-based interventions (*n* = 2). When a comparison group was included in a study, the comparison tended to be traditional ECE or a traditional playground. In these conditions, the comparison group was typically characterised by children spending more time indoors with limited time outdoors in a playground predominately comprising of manmade structures such as swings and slides. There were instances where the comparison group also received some nature-based exposure, but this exposure was still less than in the experimental group.

More detailed information on the various exposures of the included studies can be found in Appendix A. 

#### 3.2.4. Sample Size and Participant Characteristics

The total sample size of the combined quantitative and qualitative studies was 3383. Sample sizes were small across the 36 eligible studies; only three studies had a sample size greater than 200 [35,57,64], of which two were uncontrolled before and after studies [35,64] and one was controlled cross-sectional [57]. The sample size in the qualitative studies ranged from 12 [44,61] to 75 participants [45]. The age range of participants was always 2–7 years, one study included female participants only [26] and socioeconomic status (SES) was generally moderate to high, however, SES was infrequently reported.

### 3.3. Quality of Included Quantitative Studies

The quality of each quantitative study, as assessed by the EPHPP tool, can be found in Appendix A. Of the eligible studies, only two studies were of moderate quality [46,48] with the remaining rated as weak. Of the two studies rated as moderate quality, they represented the nature-based ECE (*n* = 1) [46] and ECE natural playgrounds (*n* = 1) [48] exposures. Figure 2, Figure 3 and Figure 4 present the quality rating across the quantitative studies by assessment item per outcome category. Typically, items were rated weak because of selection bias, study design, issues around confounding, and transparency over whether the researchers or outcome assessors were aware of the research questions (blinding). A weak rating for study attrition (withdrawals and dropouts) was provided in 67% of before and after studies. Data collection methods tended to be valid and reliable. 

### 3.4. Trustworthiness of Included Qualitative Studies

Figure 5 presents the findings of the trustworthiness of the included qualitative studies. On one occasion research questions (1) [58], sampling (3a) [49], data collection (3b) [44], and analysis (3c) [44] were **not** clearly described, analysis (4c) was **not** appropriate to the research question [44], and claims were **not** supported by sufficient evidence (5) [40]. It was also **unclear** if, sampling (4a) [49] and data collection (4b) [44] were appropriate to the research question on one occasion. 

### 3.5. Main Findings—Quantitative Studies

#### 3.5.1. Social and Emotional Development 

The social and emotional development domain consists of two sub-domains: social and emotional and play. Thirteen studies (representing 14 papers) included an outcome related to social and emotional outcomes: *n* = 7 nature-based ECE (*n* = 4 controlled before and after, *n* = 1 uncontrolled before and after, *n* = 1 controlled cross-sectional, *n* = 1 cross-sectional) [26,28,29,30,33,47,56,59]; *n* = 3 ECE natural playgrounds (*n* = 2 uncontrolled before and after, *n* = 1 cross-sectional) [35,48,60]; *n* = 2 natural elements within ECE (*n* = 2 cross-sectional) [50,55]; *n* = 1 garden-based interventions (uncontrolled before and after) [64].

For the play sub-domain, ten studies included a related outcome: *n* = 4 nature-based ECE (*n* = 3 controlled before and after, *n* = 1 controlled cross-sectional) [26,27,41,59]; *n* = 5 ECE natural playgrounds (*n* = 1 uncontrolled before and after, *n* = 2 controlled cross-sectional, *n* = 2 cross-sectional) [36,42,43,48,62]; and *n* = 1 types of natural element (cross-sectional) [37]. 

Findings per eligible study can be viewed in Appendix A. 

1.Social and emotional outcomes (*n* = 13 studies) (1)Nature-based ECE (*n* = 7 studies, *n* = 388 children)

Table 2 presents the results of the effect direction plot for social skills, social and emotional development, attachment (child’s ability to develop and maintain secure and positive connections with others), initiative (child’s ability to use independent thought and action), and lower behavioural problems where these outcomes were reported in more than one study. For social skills (including prosocial behaviour, social responsibility), two studies demonstrated a positive association between social skills and nature-based ECE; one was significant (*p* = 0.03, η^2^ = 0.11) [47] and the other was non-significant (mean diff: 0.32; 95% CI: −0.95, 1.59) [33]. The other study reported a positive, but non-significant association between social skills and traditional ECE (ⴄ^2^ = 0.08, *p* > 0.05) [26]. Similarly, social and emotional development was found to be positively, but not significantly associated with nature-based ECE in two studies [33,59]. However, another study demonstrated a non-significant association between nature-based ECE (*p* = 0.013) with traditional ECE (i.e., social, emotional development scores were better in children who attended traditional ECE) [56]. Findings were inconsistent in the two studies that assessed attachment [29,30,56]. One study reported that attachment scores were better in the traditional group compared to children who attended nature-based ECE at post-test (*p* = 0.058) [56] and the other study found improvements in attachment scores from baseline to follow-up in children who attended nature-based ECE [29,30]. Similarly, findings were inconsistent for initiative. One study reported that initiative scores were better in the traditional group compared to children who attended nature-based ECE at post-test (*p* = 0.187) [56] and the other study found significant improvements (*p* = 0.01) in initiative scores from baseline to follow-up in children who attended nature-based ECE [29,30]. Finally, three studies reported fewer behaviour problems in children who attended traditional ECE compared to nature-based ECE at post-test; in one study these differences were significant (ⴄ^2^ = 0.17, *p* < 0.05) [26] and non-significant in another (*p* = 0.11, η^2^ = 0.03) [47]. The other study reported fewer behavioural problems in children who attended nature-based ECE compared to traditional ECE (mean diff: −0.23; 95% CI: −1.49, 1.03) [33]. 

In addition (not reported in the effect direction plot), one study found that total protective factors (3 sub-scales: initiative, self-regulation, and attachment measured using the Devereux Early Childhood Assessment for Pre-schoolers, Second Edition) as reported by the child’s parent and teacher significantly improved from baseline to follow-up in children who attended nature-based ECE [29,30]. Another study found no association between the frequency of nature-based experiences and belief in the importance of nature-based ECE for social development [28].


(2)ECE natural playgrounds (*n* = 3 studies, *n* = 868 children)


All three eligible studies assessed outcomes related to social skills and interactions; two examined the effect of incorporating nature in the playground [35,48] and the other compared free play in ECE playgrounds with green space to indoors [60]. One study reported a significant improvement (*p* = 0.03) in social behaviour from baseline to follow-up and another reported a positive association between social interactions and free play in ECE playgrounds with greenspace [48,60]. However, another study reported significantly (*p* < 0.05) more negative interactions between teacher and child [35]. Findings also suggested that children’s emotional and behavioural outcomes, measured using the strengths and difficulties questionnaire (5 sub-scales, emotional symptoms, conduct problems, hyperactivity, peer relationships, and prosocial behaviour), significantly improved (*p* = 0.036) from baseline to follow-up [48] and stress was lower in playgrounds with green space compared to indoor free play [60].


(3)Natural elements within ECE (*n* = 2 studies, *n* = 252 children)


One study found that nature present in the ECE playground was a statistically significant predictor (regression coefficient = 0.004, *p* < 0.05) of emotional wellbeing (measured using the Leuven Well-being Scale) [50]. However, the other study reported that children’s stress levels were lower in higher quality environments (i.e., large spaces, vegetation, trees, etc.) compared to low quality environments [55].


(4)Garden-based interventions (*n* = 1 study, *n* = 336 children)


The one study utilising a garden-based intervention found a significant (all p = 0.000) and positive effect on emotional intelligence and prosocial behaviour from baseline to follow-up [64].
2.Play (*n* = 10 studies) (1)Nature-based ECE (*n* = 4 studies, *n* = 257 children)

Table 3 presents the results of the effect direction plot for play interaction, play disruption, and play disconnection in eligible studies where these outcomes were reported in more than one study. For play interaction, two studies demonstrated a positive association with nature-based ECE (i.e., play interaction was better in children who attended nature-based ECE). One study reported a mean difference of 0.86 (95% CI: −2.04–6.35) [41] and the other reported a significant difference (p < 0.001, η^2^ = 0.12) at post-test between the nature-based and traditional ECE [27,30]. However, one study found that play interaction was better in children who attended traditional ECE, but these group differences were non-significant (ⴄ^2^ = 0.11, *p* > 0.05) [26]. Findings for play disconnection and disruption were mixed with one study showing a positive association (i.e., play disruption/connection was lower in nature-based ECE) [27,30] and the other study showing a negative association) [26]. The study that demonstrated a positive association found that children at post-test in the nature-based ECE had significantly lower play disruption (*p* < 0.001, η^2^ = 0.19) and disconnection (*p* < 0.001, η^2^ = 0.12) scores compared to traditional ECE [27,30]. Whereas, the other study found that children at post-test in the traditional ECE had significantly lower play disruption (*p* < 0.001) and disconnection (*p* < 0.01) scores compared to nature-based ECE [26].

Additionally (not presented in the effect direction plot), overall play development (measured using the Kuno Beller Developmental Tables) and pretend to play (consisting of imaginative play, use of make-believe, play enjoyment, amount of emotion expressed in play, and use of make-believe in dramatic play) was higher in children who attended nature-based ECE compared to traditional ECE [26,59]. 


(2)ECE natural playgrounds (*n* = 5 studies, *n* = 347 children)


One intervention study observed children’s play behaviour at baseline and then again at follow-up after the playground was modified to include more nature elements, such as vegetation, boulders, rock, and loose parts [48]. Findings suggested improvements in playing with natural elements, risky play, solitary play, child-teacher interactions, prosocial behaviour, and fewer antisocial behaviours and lack of engagement observed in their play from baseline to follow-up [48]. Of these findings, prosocial behaviours (OR: 2.81; 95% CI: 1.17–6.91, *p* < 0.05) and playing with natural elements (OR: 7.29; 95% CI: 1.53–38.09, *p* < 0.05) significantly improved from baseline to follow-up [48]. The other eligible studies compared play in natural versus traditional playgrounds and findings across some studies indicated children engaged in more creative and imaginative play in natural ECE playgrounds [36,42,43,62]. One study reported that dramatic play was significantly higher in children who play in natural ECE playgrounds compared to manufactured ones [36]. Another study demonstrated that children in a natural ECE playground engaged in sociodramatic play for a longer duration compared to children from traditional ECE playgrounds [43]. They were also more likely to engage in object substitutions, explicit metacommunication (nonverbal cues such as tone of voice, body language, etc.), and imaginative transformations [43]. Functional and constructive play was also higher among children who played in natural ECE playgrounds compared to children who played in traditional playgrounds, but creative and imaginative play was lower [42]. However, another study noted functional and imaginative play was higher among children who played in traditional ECE playgrounds compared to children who played in natural ECE playgrounds [62].


(3)Natural elements within ECE (*n* = 1 study, *n* = 36 children)


One study measured cognitive play (consisting of functional, constructive, exploratory, dramatic, games with rules) across three different playground types: natural, mixed, and manufactured) [37]. The authors found that the natural area afforded greater dramatic, exploratory, and constructive play compared to the mixed and traditional zones [37]_._

#### 3.5.2. Cognitive Development

Eleven studies (representing fifteen papers) presented an outcome related to cognitive development: *n* = 7 nature-based ECE (*n* = 5 controlled before and after, *n* = 1 controlled cross-sectional, *n* = 1 cross-sectional) [26,27,28,29,30,31,32,33,47,56,59]; *n* = 1 ECE natural playgrounds (cross-sectional) [60]; *n* = 1 natural elements within ECE (cross-sectional) [53]; and *n* = 2 garden-based interventions (uncontrolled before and after) [38,64]. 

Findings per eligible study can be viewed in Appendix A.
3.Cognitive (*n* = 11 studies)(1)Nature-based ECE (*n* = 7 studies, *n* = 438 children)

Table 4 presents the results of effect direction plot for attention and self-regulation in eligible studies where these outcomes were reported in more than one study. For attention, two studies demonstrated a positive, but non-significant association with children who attended nature-based ECE [27,30,33], and one study demonstrated a non-significant negative association (i.e., children in the traditional setting had better attention scores) [47]. Across three studies, there was a positive association between self-regulation and nature-based ECE. Two studies demonstrated a significant association [29,30,47] and one demonstrated a non-significant association [56]. 

There were a number of additional findings not reported in the effect direction plot because outcomes could not be grouped together. At post-test, one study reported that there was a small and non-significant effect on working memory (*p* = 0.19, η^2^ = 0.02) and a non-significant effect on inhibition (*p* = 0.76, η^2^ = 0.00) between the intervention and control group [47]. There was no significant differences between the nature-based ECE and control groups for overall executive function score (*p* = 0.60, η^2^ < 0.01) [30,32]. In another study, cognitive development was lower and teacher perception of language development was higher in children who attended nature-based ECE, however, the differences between the nature-based ECE and control groups were non-significant [59]. One study reported that there were no significant differences in the communication scores at post-test for children who attended nature-based ECE compared to the control group (p = 0.694) [56]. At post-test, total learning behaviours (consisting of attention, competence motivation and attitudes) was higher in children who attended nature-based ECE compared to traditional ECE, but this was non-significant (*p* = 0.12, η^2^ = 0.02) [27,30]. Kindergarten readiness (counting, rhyming, recognition) was lower in children who attended nature-based ECE compared to the control, but these differences were non-significant (*p* > 0.05, ⴄ^2^ = 0.16) [26]. There were non-significant differences in curiosity scores in children who attended nature-based ECE compared to the control group [30]. Finally, there were significant improvements (all *p* < 0.001) in creativity (consisting of fluency originality, and imagination) from baseline to follow-up in children who attended nature-based ECE [31]. 


(2)ECE natural playgrounds (*n* = 1 study, *n* = 16 children)


The one eligible study in this exposure category assessed visual spatial scores (an indicator of children’s direct attention) to determine if there were differences in children who had engaged in free play in ECE playground with green spaces compared to children who were indoors [60]. It was found that children who had been exposed to ECE playgrounds with green space had higher visual spatial accuracy scores compared to the control [60].


(3)Natural elements within ECE (*n* = 1 study, *n* = 198 children)


One eligible study in this exposure category assessed attention in children who attended ECE setting with high-quality environments (i.e., large spaces, vegetation, trees, etc.) versus those who had a low-quality environment [53]. The authors found that the two domains of attention: hyperactivity (*p* = 0.069) and inattention (*p* < 0.05) were associated (i.e., better) in schools with high-quality environments, and inattention was significantly associated [53]. 


(4)Garden-based interventions (*n* = 2 studies, *n* = 391 children)


One study reported significant improvements (*p* < 0.01) in all sub-categories of scientific from baseline to follow-up (47). The other study reported that delay gratification (self-regulation) and visual motor integration did not significantly improve from baseline to follow-up [38].

#### 3.5.3. Nature Connectedness 

Nine studies included an outcome related to nature connectedness and all nine studies were included in the nature-based ECE category, of which, *n* = 3 were controlled before and after [46,47,59], *n* = 2 uncontrolled before and after [63,65], *n* = 3 controlled cross-sectional [34,54,57] and *n* = 1 cross sectional [28].

Findings per eligible study can be viewed in Appendix A.
4.Children’s connectedness to nature (*n* = 9 studies)(1)Nature-based ECE (*n* = 9 studies, *n* = 792 children)

Table 5 presents the results of effect direction plot for nature relatedness/biophilia, environmentally responsible behaviour, and awareness of nature/environment in eligible studies where these outcomes were reported in more than one study. For nature relatedness (or biophilia), five studies were positively associated with nature-based ECE [46,47,54,57,65]. Four of these studies demonstrated a significant association [46,54,57,65] and one demonstrated a non-significant association [47]. One study demonstrated no difference in nature relatedness scores between nature-based and traditional ECE [34]. For environmentally responsible behaviour, two studies demonstrated a positive, but non-significant association with traditional ECE (i.e., environmentally responsible behaviour was better in children who attended traditional ECE) [46,47]. However, another study reported that environmentally responsible behaviour was significantly better in children who attended nature-based ECE compared to traditional ECE [57]. Finally, in two studies, awareness of nature was positively associated with children who attended nature-based ECE compared to children who attended traditional ECE) [57,59]. 

Additionally (findings not reported in the effect direction plot), there were improvements in knowledge and skills of nature in children who attended nature-based ECE [63] and awareness of the surrounding environment was also higher in children who attended nature-based ECE [59]. 

### 3.6. Main Findings—Qualitative Studies (n = 10 Studies)

Ten studies were included in the thematic analysis, of which, six studies involved nature-based ECE, three studies were ECE natural playgrounds, and one was natural elements within ECE (study characteristics of qualitative studies can be found in Appendix A). Studies tended to use direct observation and interviews (predominately with educators) to collect qualitative data. Findings from the thematic analysis are presented in Figure 6 and show four higher order themes. 

**Theme** **1.**
*Natural settings provide more affordances compared to traditional settings.*


Theme 1 indicated the importance of the natural environment for affording opportunities to enhance a range of outcomes. This theme included seven subthemes relating to the different affordances that nature provides compared to traditional settings. Seven of the included studies noted that the natural environment enabled children to diversify their play (subtheme 1.1), such as imaginative, risky, exploratory, and active play [37,39,40,44,45,52,61]. Diversifying play is not only important for children’s total movement but also their social interactions, creativity, and learning. The importance of play can be described in the following quote.


*“The children also invent themselves; when they have stimulus for their eyes, children invent it [activity] without your help. And it should be like this; some part should be like this. But you need to have stimulus. It’s not enough to have a brown yard and a climbing frame. So, it [green yard] added somehow; they definitely had good games. They pretended that they had a campfire, they got the stones as sand pretended that they were on a trip. And their imagination was in use there, and when children use their brains, natural tiredness arises, and it did them good, a lot of good. Then rest comes naturally, and you have a good appetite and we’re in the positive cycle. So they could use their imagination, and we encouraged them. We didn’t prohibit them, we just advised them not to rip anything.”*
[61]

The next subtheme (1.2) highlighted that in two studies natural settings afforded children with higher levels of risk compared to traditional settings [49,52]. This relates to the above subtheme as risky play is a type of play important for children’s development. Risky play is characterised by play that is “thrilling and exciting and where there is a risk of physical injury” [52]. Although this may seem potentially dangerous to children, this type of play is important for children’s development [66]. 


*“*
*I like playing in the fallen logs and trees on the playground; it is so much fun, but a bit scary too! I like the big pile of sticks and logs that we made—it is for another fort that is going to be really high off the ground.”*
[49]

Similarly, related to subtheme 1.1, three studies also noted that natural settings afforded more variation (the space and elements) to support children to use and increase their imagination and creativity (subtheme 1.3) [37,39,49].


*“I like being outside with my friends. We make shelters and we make up different games, like getting trapped on an island, or being on a boat and making our escape! I like doing science outside too—like different experiments, especially when the sun is out.”*
[49]

The fourth subtheme (1.4) relates to the importance of social interactions. Four studies demonstrated that natural settings enabled peers and teachers to have prosocial interactions in relation to encouraging play [39,44,49,51].


*“The children are shouting ‘X… can’t you catch us? Please catch us, try to catch us …’. The staffs join the situation and run after the children. The children are shouting ‘Catch me … can’t catch me’ … There is excitement and the staff are running after the children, catching them and holding them before releasing them. The staff have high energy, the children focus on the adults, avoiding being caught. The adults show empathy, holding and hugging the child when it is caught. The game is exciting and creates enthusiasm. A high level of physical activity is created, by climbing up, sliding down, running around and hiding in the tower to escape capture by the adults. They run at high speed and the children’s body language shows that they are very much engaged in the game.”*
[51]

Three studies highlighted that natural settings increased child-initiated learning and students perceiving themselves as capable learners compared to traditional settings (subtheme 1.5 and 1.6) [37,44,58]. 


*“[CogG] has poor concentration, sees herself as the baby, finds it difficult to sit and listen to story. She is extremely lacking in confidence … shy … she won’t look at you indoors. With child-led learning she is totally engrossed and remains on task. Outside is the best learning environment for her … she remains on task. When outside she will come over and say ‘I like this’ and ‘I like doing that’, ‘this is my favourite place’.”*
[58]

Finally, three studies highlighted that children increased contact with nature enabling them to increase their knowledge of nature (subtheme 1.6) [39,44,61].


*“Especially about the forest floor mat, I remember that our children kept asking, ‘what is it’ and ‘what’s growing there’ and explored it very carefully; they were almost lying on their stomachs there. Especially the older ones, and they had a lot of questions about it.”*
[61]

**Theme** **2.**
*Natural and traditional settings provide similar affordances.*


Despite Theme 1 indicating the importance of natural affordances for a range of outcomes, some studies also reported that natural and traditional settings provided similar affordances. The one subtheme noted that the opportunity for and frequency of risky play was similar in both natural and traditional ECE settings [52]. This subtheme relates to the findings reported on risky play in subtheme 1.2. Taken together, children will seek risk irrespective of playground type, however, the natural environment affords greater risk (Theme 1, subtheme 1.2) [52].


*“Comparing the two play environments, they both seem to include an extensive number of affordances for risky play. At both preschool playgrounds, there are opportunities for play in great heights such as climbing, jumping down, and balancing and as well as opportunities for play with high speed such as swinging, sliding/sledding, running, and bicycling.”*
Taken from authors conclusions [52]

**Theme** **3.**
*Children’s preferences of setting types.*


Two studies reported that the natural environment is more diverse and engaging and preferred by children compared to traditional settings (subtheme 3.1) [49,51]. It appears when children were outdoors in nature it afforded them the opportunity to play in a diverse environment with their friends and this combination provided enjoyment. 


*“I like going outside and playing! I like playing with my friends, Sydney and Megan. We play hide and seek on the playground and hide in the forest in the logs and trees. I like outside because it’s so fun and I really like to play. Sometimes I play with my sister too; I like all the colours outside and all the space.”*
[49]

However, another study suggested that mixed areas (combining both natural with traditional elements) were preferred by children [37]. 

**Theme** **4.**
*Restorative and invigorating effect of nature.*


Two studies indicated the importance of the natural playground in helping children invigorate and/or restore their energy for the diverse types of play children engage in [39,61]. For example, the natural environment for some children provided them with more energy to continue playing, however, other children may feel the requirement to nap, thus restoring their energy to engage in more play [39,61].


*“Now it’s become very difficult to finish playing. They would rather continue, and those who need to take a nap, they’ve had a nice, long time outdoors and nice games, so they fall asleep more easily, and it affects their energy in the afternoon. Some children have very long days here. They come in the morning and stay until five o’clock; they seem to be somehow energetic and lively in the yard. This is new for us. The contrast to the previous yard is so great that the effects can be seen here very quickly.”*
[61]

### 3.7. Synthesis of Quantitative and Qualitative Findings

Of the outcomes assessed in quantitative studies, initiative, behavioural problems, play disruption and play disconnection did not emerge as themes from the qualitative studies. Appendix A shows the matrix relating themes from the qualitative evidence synthesis with the findings from the quantitative evidence synthesis. The matrix indicates where findings from the two data sources were confirmatory or conflicting. Themes not presented in the matrix could not be directly linked to the results of the quantitative synthesis. However, these themes were considered for generating hypotheses on how or why observed quantitative results occurred.

### 3.8. Social, Emotional and Environmental Development

**Social and emotional.** From the quantitative studies, findings suggested that children who attended nature-based ECE had improved social skills and social and emotional development. From the qualitative synthesis, this might be achieved through three possible mechanisms: children diversify their play, they have increased creativity and imagination, and engage in prosocial interactions with peers and teachers. When children diversify their play, those who attended nature-based ECE have more play interactions in comparison to traditional ECE which could facilitate the acquisition of greater prosocial skills. Through sociodramatic and symbolic play, children create and share common narratives with their peers, acting out and enacting imaginary situations, which may play an important role in fostering imagination and creativity, understanding social complex structures, and improving social skills. The quantitative analysis suggested improvements in children’s social skills, interaction, and development. However, it was unclear whether children had improved attachment (a child’s ability to promote and maintain positive connections with others), and one study noted negative child-teacher interactions in children who engaged in natural playgrounds. See Figure 7 for an illustration of the pathway on how nature-based ECE could influence social skills.

**Play.** From the quantitative synthesis, children who attended nature-based ECE had more play interactions in comparison to traditional ECE. This is supported in the qualitative synthesis as play interaction might be achieved through children diversifying their play, increased risk, increased creativity and imagination, and prosocial interactions with peers and teachers. It was noted that natural settings enable children to engage in a diverse range of play types, including risky play, solitary play, dramatic play, sociodramatic play, functional and constructive play, etc. These diverse play types will provide continuous opportunities for children to enhance their play interactions. Similarly, it was noted that nature-based ECE provides children with higher levels of risk, which enable risky play, a type of play that plays a pivotal role in managing health and wellbeing. As mentioned previously, sociodramatic and symbolic play where children may act out imaginary situations with their peers also improved imagination and creativity. Finally, peers and teachers may facilitate play opportunities through a wider range of opportunities (affordances) provided by natural settings. See Figure 8 for the pathway on how nature-based ECE could influence play interaction.

### 3.9. Cognitive Development

From the quantitative studies, findings suggested that self-regulation (ability to understand and manage behaviour) is better in children who attend nature-based ECE. The qualitative evidence raised potentially important factors and processes that may help explain the outcomes: children diversify their play, engage in higher levels of risk, see themselves as capable learners, and have prosocial interactions with peers and teachers. Through diverse types of play, including risky, sociodramatic, and symbolic play, children learn to negotiate through different situations, experiment individually and collectively different ways to solve problems, which in turn creates a playful disposition of self-regulation. Similarly, when children see themselves as capable learners by developing different cognitive skills through nature-based ECE it might be linked to self-efficacy and self-confidence, which in turn may foster self-regulation. See Figure 9 for the pathway on how nature-based ECE could influence self-regulation. It is important to note that self-regulation is a complex construct, influenced by many different factors. The pathway presented in Figure 9, may explain some possible causal pathways of self-regulation, pertinent to nature-based ECE. However, the likelihood is that there will be a multitude of other factors influencing self-regulation that are beyond the scope of this review.

### 3.10. Nature Connectedness

Evidence from the quantitative studies suggested that children who attended nature-based ECE had higher levels of biophilia and awareness/knowledge of nature. Using evidence from the qualitative synthesis, this might be achieved through four possible mechanisms: children diversify their play, they have increased contact with nature, child-initiated learning, and capable learners, and peers and teachers have prosocial interactions. Through engaging in different types of play in natural settings, children are exposed to different natural elements that they could use in their play which in turn might increase connection with and awareness of nature. Children seeing themselves as capable learners where they develop different cognitive skills such as attention, and where children take the lead on what they want to learn may also improve biophilic tendencies and awareness of nature by using and learning about the natural elements at their disposal. Finally, peers and teachers can help support awareness of nature and biophilia by enabling children to explore the natural environment. See Figure 10 for the hypothesised pathway on how nature-based ECE influences biophilia and awareness of nature.

## 4. Discussion

This systematic review aimed to understand whether nature-based ECE is associated with children’s social, emotional, and cognitive development. Based on very low certainty of evidence, findings from the effect direction plot indicated positive associations between nature-based ECE and children’s self-regulation, social skills, social and emotional development, nature relatedness (or biophilia), awareness of nature, and play interaction. There were inconsistent associations between nature-based ECE and children’s attention, attachment, initiative, environmentally responsible behaviour, and play disruption and disconnection. Finally, there was a negative association between nature-based ECE and children’s behavioural problems (i.e., there were higher behavioural problems in children who attended nature-based ECE). The qualitative synthesis noted that nature-based ECE afforded opportunities for children to engage in diverse types of play, use their imagination and creativity, have prosocial interactions, and have increased contact with nature in comparison to traditional ECE.

Health outcomes of attending nature-based ECE are likely to be impacted through a number of plausible mechanisms. Nature-based ECE settings are inherently unique and exposure to nature alone could impact certain child health outcomes directly, or indirectly through interconnecting mediators; for example, the rich and diverse natural environment may provide affordances [17] for children to engage in a range of play types that facilitate social interaction, creativity and imagination and/or physical activity which may impact a range of health outcomes. These possible pathways have been presented in the mixed-synthesis of this systematic review and could provide researchers with future research questions in relation to nature-based ECE.

One such pathway, which is the common factor across all outcomes featured in this review, is play. It is suggested that nature-based ECE provides children with an immersive nature experience where children interact with different natural spaces and elements that afford opportunities to engage in different types of play. In a recent systematic review of nature play on children’s health and wellbeing, the authors reported increases in six different types of play (functional, constructive, exploratory, dramatic, imaginative, and symbolic) in nature settings compared to a comparison [18]. It is generally agreed that engaging in different types of play has an impact on children’s development across several outcome domains [67,68]. However, limited research exists on the plausible mechanisms by which different types of play improve different health and development outcomes in childhood [67]. For example, it might be that when children play, they are engaging in physical activity and physical activity improves several childhood health outcomes. However, not all play is physically active, and even play which is sedentary in nature (e.g., sociodramatic and symbolic play) could have improvements in other outcomes. Furthermore, other mediators could feature on the pathway, such as play facilitating social connections with other children. The potential added benefit of nature-based ECE is that exposure to nature could afford greater opportunities for engaging in diverse types of play. For example, in the same review noted above [18], the authors also highlighted that nature-play was also likely to improve cognitive, social, and emotional related outcomes. The findings are similar to the present study, however, the authors did not make any reference to the possible mechanisms by which outcomes may have occurred [18].

As previously noted, it is likely that nature-based ECE affords greater opportunities for play, which facilitates social interactions. This review highlighted that both social skills and social and emotional development were positively associated with nature-based ECE, however, inconsistent associations were found for attachment and initiative. Although few studies have explored the relationship between nature-based ECE and social and emotional outcomes, there are several conceptually similar systematic reviews that have synthesised the impact of nature on these outcomes. Dankiw and colleagues’ review supported the position that nature encouraged a diversity of play, where three out of four included studies demonstrated improvements in aspects of social outcomes [18]. Similarly, both studies included in their narrative synthesis noted improvements in domains of emotional outcomes [18]. Another systematic review of the effect of exposure to nature (including non-ECE settings) and children’s social and emotional development found less favourable findings for establishing and maintaining relationships [12]. From seven observational studies and four experimental studies, a total of 36 analyses were included in the synthesis, of which, 19.4% found positive associations between nature and children’s ability to maintain relationships [12]. Despite these efforts to understand the impacts of nature on children’s social and emotional outcomes it remains unclear what the plausible pathways are that impact children’s social and emotional outcomes, and the interconnecting role of play and social development on other physical and cognitive outcomes. One hypothesis presented in the mixed-synthesis is that nature-based ECE may provide specific construct, composition, and quality that affords children opportunities to engage in diverse types of play. Through play, greater social connections are facilitated as when children play they must cooperate with other children, solve problems and create rules for their play [69]. Problem solving during play may impact short-term social and emotional skills, such as empathy, attachment, and emotional flexibility [69,70]. Understanding the role of nature-based ECE in facilitating the unique dynamics in children’s play may be important to understanding children’s longer-term social and emotional development as well as other related outcomes.

One such outcome that could be impacted by the interaction between nature, play, and social connections is self-regulation. The mixed synthesis suggested that nature-based ECE settings could develop self-regulation through a number of plausible mechanisms, including diversifying play (particularly free play, active play, risky play, and sociodramatic play) [67] and social interactions. Self-regulation relates to the child’s ability to understand and manage behaviour and predicts a child’s social, emotional, and behavioural development [71]. During play, children have control over their own activity, set challenges, and regulate their own behaviour, and the key features of play (uncertainty, flexibility, novelty, and open-ended) create the conditions for the development of self-regulation [67,72]. In the present study, based on very low certainty of evidence, self-regulation was positively associated with children who attended nature-based ECE. In other words, children who attended nature-based ECE demonstrated better self-regulation than those who attended traditional ECE. These findings have been echoed in conceptually similar fields. In a study that aimed to explore whether active play during recess was associated with children’s (*n* = 51, 4.8 years) self-regulation and academic achievement [73], the authors noted a positive association between active play and self-regulation (*β* = 0.43, *p* = 0.001), and that self-regulation mediated the relationship between active play and academic achievement (math *β =* 0.18, *p* = 0.03; emergent literacy *β* = 0.20, *p* = 0.035) [73]. Although supportive of our hypothesised pathway, design issues (e.g., small sample size and cross-sectional design) mean we should be cautious when inferring causality. Children being exposed to nature could also have an impact on self-regulation development. In a quasi-experimental study, the authors aimed to explore whether the frequency and duration of “green” schoolyards impacted children’s self-regulation development [74]. Findings indicated that higher frequency and duration in green schoolyards were associated with greater improvements in self-regulation [74]. However, the frequency was significant for girls only and duration was significant in autumn for females only [74]. The effect of nature-based ECE on children’s cognition, through self-regulation, is an area that may need greater research emphasis to understand the pathways in which cognitive outcomes are likely to improve.

As mentioned, nature-based ECE is likely to be play-promoting in that the rich, diverse natural space and elements are likely to afford different types of play. Engaging in different types of play will also encourage more social interactions and physical activity; this might be one pathway to improving cognitive outcomes. Alternatively, simply being in nature, particularly of high frequency and duration [70,74], may in itself enhance self-regulation through its interdependencies with attention. Self-regulation is a complex construct that branches across numerous fields of psychology. It can relate to both the conscious and automatic processes in which individuals monitor, manage, and control their behaviours, thoughts, emotions, and interactions with the environment, including task performance but also including social interactions [75]. Importantly, underpinning a child’s ability to self-regulate is their capacity to maintain focused attention [76]. More specifically, we can draw upon the work of Stephen and Rachel Kaplan and their insights into the restorative benefits of nature, including its capacity to restore directed attention and improve concentration [77,78]. The natural world provides young children with the ability to separate themselves from the other environments in their lives, it provides stimulating, absorbing, immersive, and enjoyable stimuli that can hold one’s attention without any effort being expended (e.g., a fascination with insects, sticks, or water). With these factors being present to varying degrees in nature-based ECE, the restorative benefits of nature may afford young children the optimal conditions to develop their self-regulatory processes [74,79]. Despite these possible links between nature, self-regulation, and attention, the present study noted inconsistent associations between nature-based ECE and attention based on very low certainty of evidence. Similarly, a systematic review of exposure to nature and children’s (conception—12 years) socioemotional development found that 17% of analyses from observational studies had positive associations with attention and 50% reported positive associations in experimental studies [12] which echo the inconsistent findings of this study. This might also indicate the need for more robust experimental studies to optimise the chance of accurately detecting the impact—if one exists—of nature-based ECE on children’s attention and other cognitive outcomes.

There were also a number of outcomes related to children’s connectedness to nature, including nature relatedness and awareness of nature that were positively associated with nature-based ECE. Inconsistent associations were found for environmentally responsible behaviour. Most of the eligible studies measured outcomes over a short period, so it might be the awareness and nature-relatedness outcomes are likely to produce change over a short period of time. Whereas environmentally responsible behaviour, which perhaps relates more to knowledge is a longer-term outcome and requires longitudinal exploration using research designs such as cohort studies (prospective and retrospective).

### 4.1. Strengths and Limitations of the Review

This is the first comprehensive systematic review of both quantitative and qualitative evidence that aimed to understand the association between nature-based ECE and children’s social, emotional, and cognitive development, and children’s, parent’s, and/or practitioner’s perceptions of nature-based ECE on these outcomes. This systematic review was registered to PROSPERO in October 2019 and subsequently, a peer reviewed protocol was published to BMC Systematic Reviews [10]. Throughout each step of the systematic review, a steering group comprised of experts from policy, research, and practice was consulted to ensure both rigour and relevancy of the review process and findings. We conducted a thorough search for research by searching nine databases, websites, contacting relevant national and international stakeholders, dissertations, and theses, and there were no restrictions on publication year or language. Finally, we aimed to ensure the rigour of this review by following the recommended systematic review procedures; both full-text articles and study quality was assessed independently by two reviewers and one reviewer completed data extraction which was checked by a second reviewer.

Despite the numerous strengths of this review, there were limitations that were mitigated accordingly. For example, the large number of articles retrieved meant that title and abstract screening could not be completed in duplicate, however, a second reviewer checked 10% to minimise errors. We also excluded studies that recruited solely children with a chronic disease, mental health condition, and or/learning disability as these populations were beyond the scope of this review. However, there is a theoretical case that such populations might benefit even more from nature-based ECE, and this should be synthesised in a future review. Minor modifications to the EPHPP tool to define the target population, specify confounders of interest, and enhance the overall rating of the paper were made to ensure it was relevant for the current review. Although unlikely, we cannot be sure whether these small modifications of words and terms impacted the validity and reliability of the EPHPP tool. In addition, despite these minor modifications, the EPHPP tool rated most of the eligible studies as weak, particularly across the selection bias, blinding, and attrition (before and after studies only) domains. As noted in a conceptually similar systematic review looking at immersive nature experiences on children’s health, it is important to highlight that in this field these tools (EPHPP and others) which assess study quality may provide weak ratings in certain domains, e.g., blinding, where it is difficult to conduct research practices that are the gold standard [11]. Finally, we assessed the certainty of evidence using GRADE as recommended, however, the authors recognise the limitation of using this tool as it is unlikely that randomised controlled trials (RCT) could be used in this field to evaluate the effect of nature-based ECE on child health outcomes [80]. GRADE attaches more weighting to RCTs designs, with all other study designs starting at a rating of low; meaning that there is no variation in ratings between the eligible studies despite some of these studies reflecting the “best available evidence” in this field.

### 4.2. Strengths and Limitations of the Evidence

Thirty-six studies reported social, emotional, and cognitive development, of which, 12 studies used an experimental design (6 uncontrolled before after; 6 controlled before and after) and consisted of a good geographical spread across most continents. Studies also tended to use valid and reliable measures for assessing outcomes (data collection) and qualitative studies (*n* = 10) demonstrated trustworthiness. However, most studies were cross-sectional in design (*n* = 15) and tended to have small sample sizes, both of which limit our ability to draw conclusions on the findings. Of the 27 eligible quantitative studies (including one mixed methods), only two studies were rated moderate, and the remaining were rated weak. In most instances, weak ratings were provided because of study design (cross-sectional or controlled cross-sectional), selection bias, blinding, confounders and attrition. Finally, no studies were conducted in low-income countries.

### 4.3. Future Directions

To improve the evidence base and thus enhance our ability to draw stronger conclusions, research must improve the study quality, description of exposure, and longitudinal impacts.

All but two studies were rated weak. The reasons for weak rating were described previously, but predominately centred on eligible studies primarily being of cross-sectional design, which were rated weak for that domain according to the EPHPP tool. Research efforts must move from cross-sectional designs to more robustly designed experimental studies with control groups that are adequately powered. This would have significant implications for the field and help researchers understand the complex mechanisms in which nature-based ECE improves childhood health and development.

Simultaneously, if efforts are made to support robust study designs as outlined above, a thorough description of the nature exposure, such as time, frequency of visits, and quantifying nature needs to be included [81]. Nature potentially impacts several health outcomes across a number of possible mechanisms, by describing the extent of nature exposures within nature-based ECE will support the identification of the specific pathways that nature-based ECE is likely to impact social, emotional, and cognitive development [81]. This will enable the field to understand how much nature is required within nature-based ECE settings to produce an impact on developmental outcomes beyond children’s normative developmental trajectories.

None of the eligible studies assessed the longitudinal impacts of attending nature-based ECE despite the likelihood that nature-based ECE would have longitudinal impacts on social, emotional, and cognitive outcomes beyond the developmental norm for children 2–7 years. Longitudinal studies are critical as they enable the field to understand the possible mechanisms in which outcomes would improve, the timing and degree of improvements and/or harms, and whether improvements are sustained as children transition into primary/elementary education where they are likely to see a reduction of exposure to nature.

## 5. Conclusions

Findings from this systematic review suggested, based on low certainty of evidence, that there were positive associations between nature-based ECE and children’s self-regulation, social skills, social and emotional development, nature relatedness (or biophilia), awareness of nature, and play interaction. There were inconsistent associations between nature-based ECE and children’s attention, attachment, initiative, environmentally responsible behaviour, and play disruption and disconnection. A negative association between nature-based ECE and children’s behavioural problems was found.

This systematic review hypothesised that a nature-based ECE, with a rich and diverse environment, affords children the opportunities to engage in diverse types of play which is likely to impact social, emotional, and cognitive developmental outcomes. To test this hypothesis and understand more about the mechanisms in which nature-based ECE impacts children’s social, emotional, and cognitive development, the evidence base must move to higher quality study designs that adequately describe the nature exposure more precisely and with robust methods and over a longer duration. This will elevate the current evidence base and inform research, policy, and practice on the complex pathways in which nature-based ECE impacts children’s social, emotional, and cognitive development.

## Figures and Tables

**Figure 1 ijerph-19-05967-f001:**
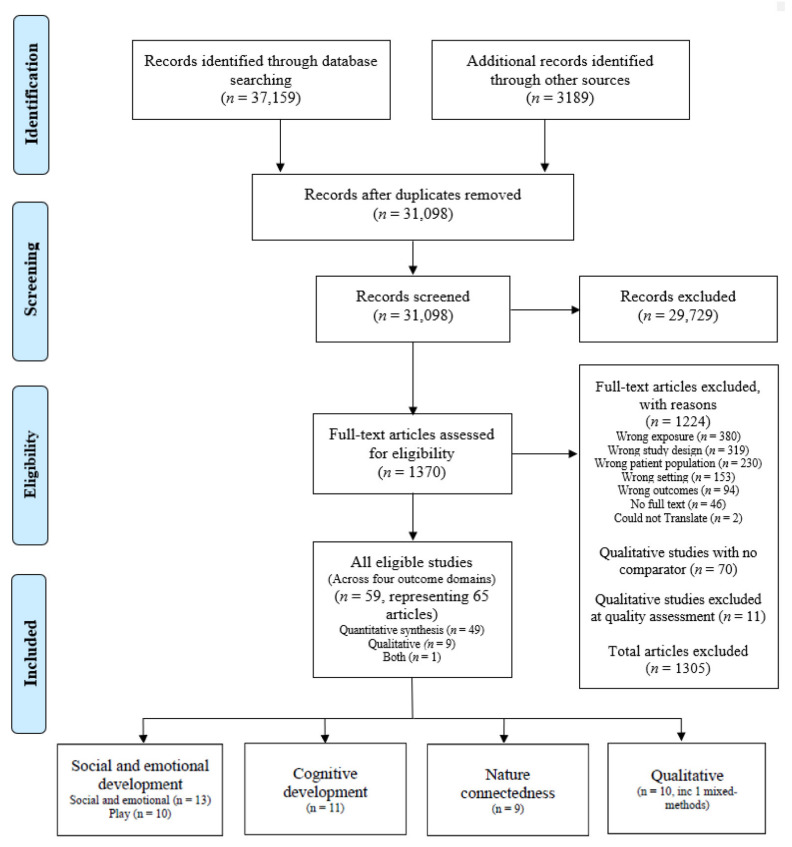
Results from the literature search.

**Figure 2 ijerph-19-05967-f002:**
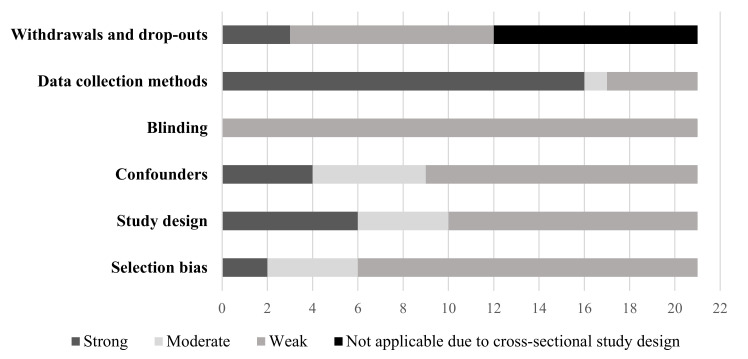
Quality of quantitative studies by assessment item—Social and Emotional Development.

**Figure 3 ijerph-19-05967-f003:**
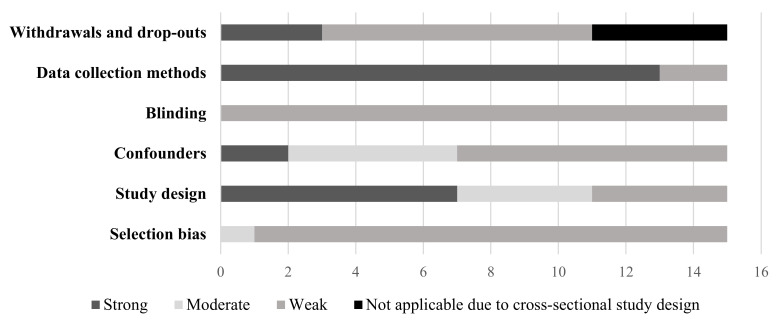
Quality of quantitative studies by assessment item—Cognitive Development.

**Figure 4 ijerph-19-05967-f004:**
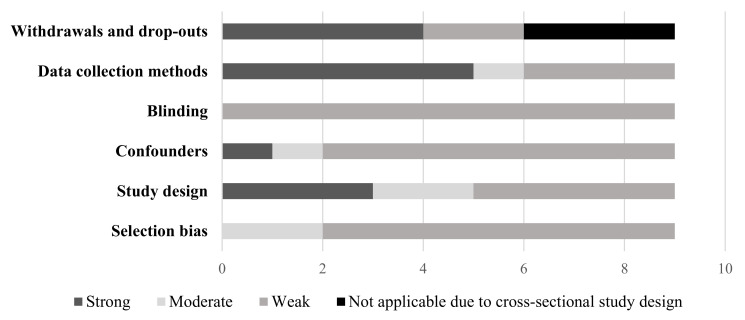
Quality of quantitative studies by assessment item—Nature Connectedness.

**Figure 5 ijerph-19-05967-f005:**
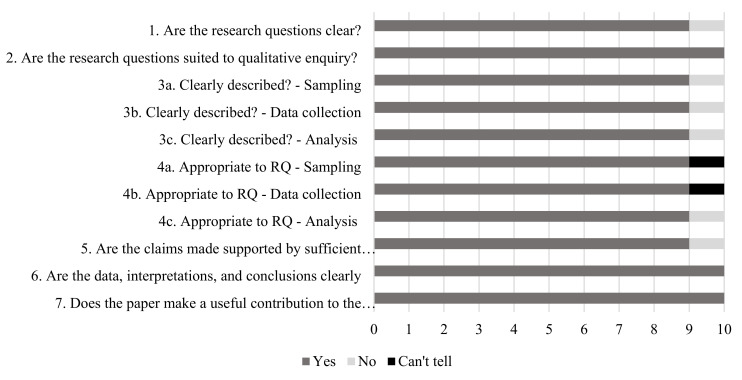
Trustworthiness of qualitative studies by assessment item.

**Figure 6 ijerph-19-05967-f006:**
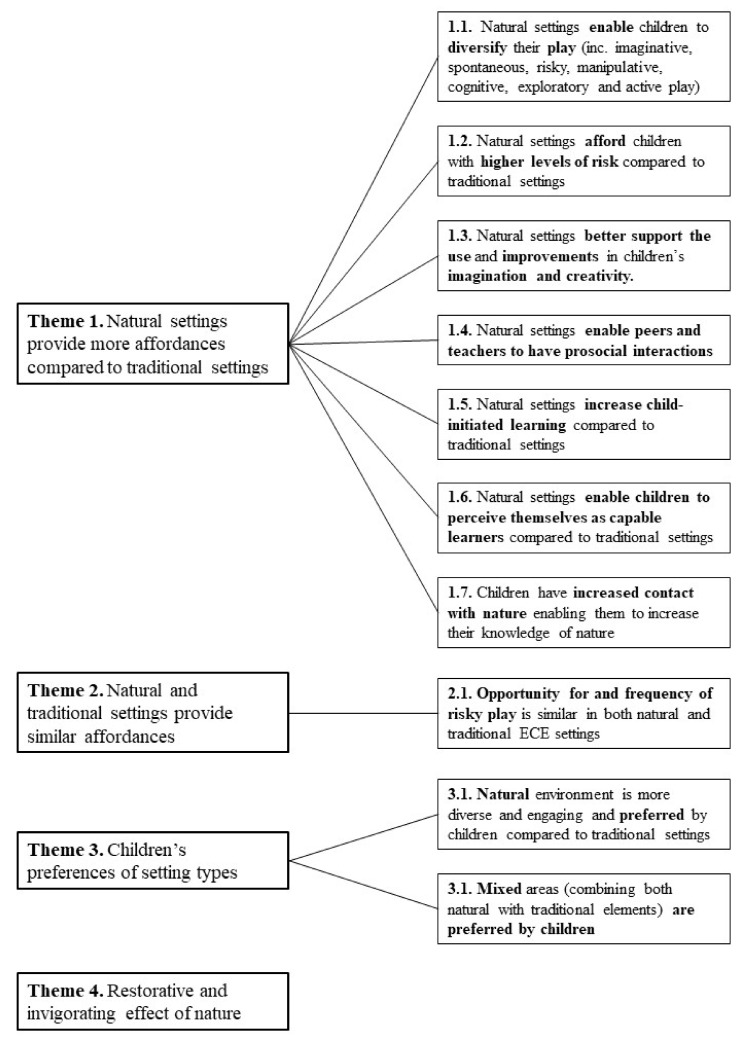
Findings from the thematic analysis.

**Figure 7 ijerph-19-05967-f007:**
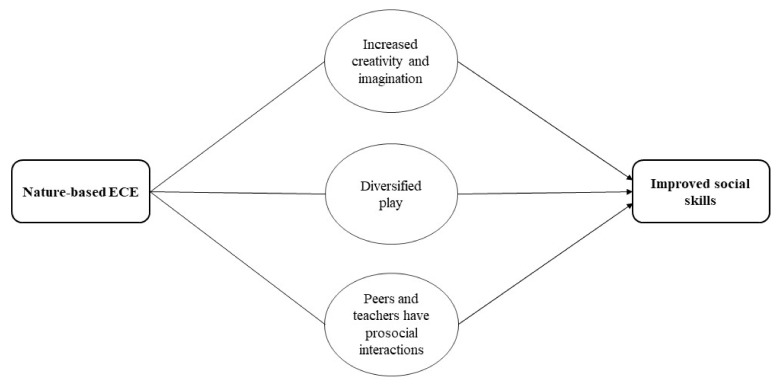
Convergent synthesis on how nature-based ECE could influence social skills. An arrow denotes where factors lead to an influence on social skills.

**Figure 8 ijerph-19-05967-f008:**
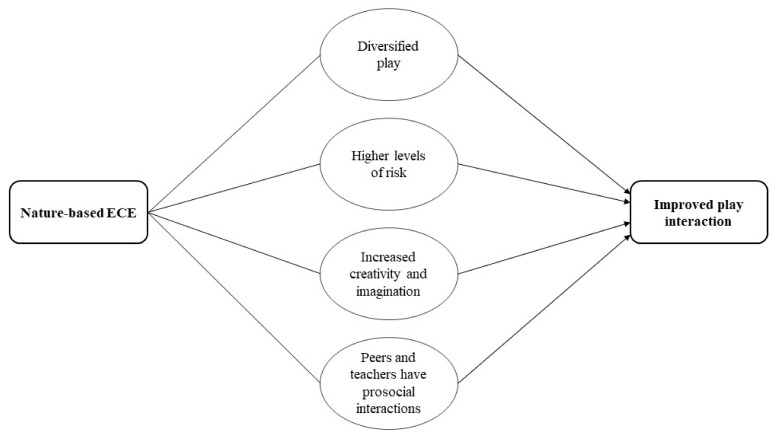
Convergent synthesis on how nature-based ECE could influence play interaction. An arrow denotes where factors lead to an influence on play interaction.

**Figure 9 ijerph-19-05967-f009:**
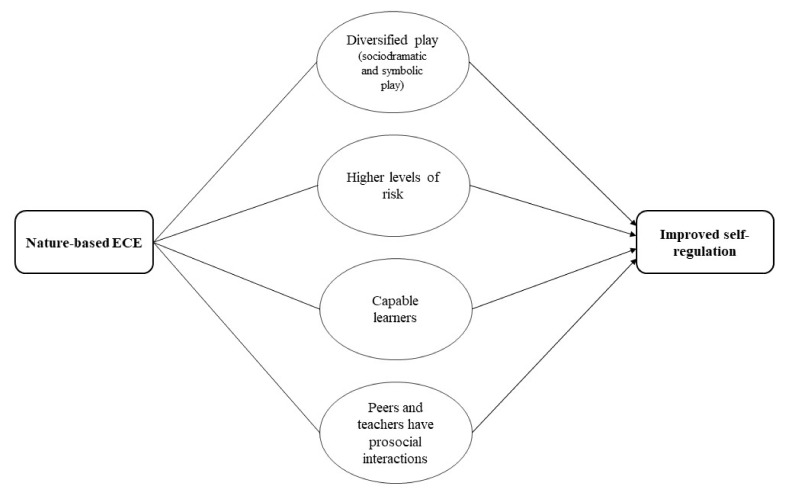
Convergent synthesis on how nature-based ECE could influence self-regulation. An arrow denotes where factors influence self-regulation.

**Figure 10 ijerph-19-05967-f010:**
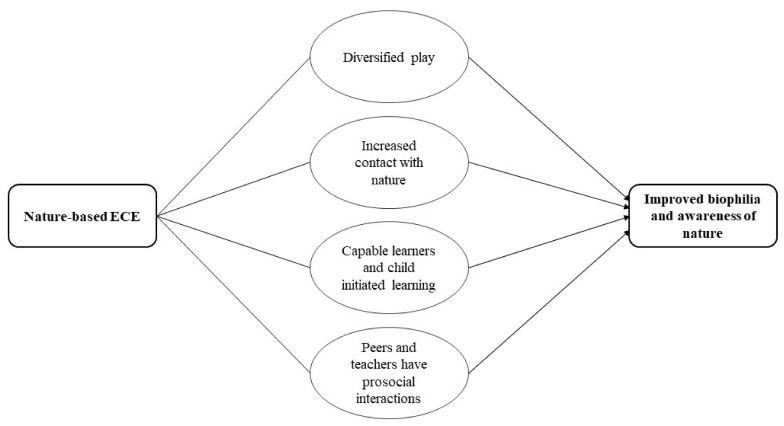
Convergent synthesis on how nature-based ECE could influence biophilia and awareness of nature. An arrow denotes where factors lead to an influence on biophilia and awareness of nature.

**Table 1 ijerph-19-05967-t001:** Overview of the exposure categories.

Nature-based ECE	This category represents studies with a higher exposure to nature. These ECE settings would integrate rich and diverse natural elements in their environment and children would spend most of their ECE time outdoors. Examples of a typical nature-based ECE environment may include: wooded areas, forest, trees, hills etc. ECE practitioners would be present and may lead on formal and informal educational activities that involve/incorporate nature.
ECE natural playgrounds	This category includes studies that utilize interventions that enhanced the nature in the playground or where natural playgrounds were compared to traditional playgrounds. Children would typically spend less time outdoors in nature in these studies.
Natural elements within ECE	This category represents a lower exposure to nature and included studies (mostly cross-sectional in design) that looked at the association of specific natural elements, such as trees, vegetation, hills, grass, etc., or specific features or quality of the playground on specific health outcomes.
Garden-based interventions	This category represents studies that included an intervention with a garden component and was delivered within an ECE setting.

**Table 2 ijerph-19-05967-t002:** Effect direction plot for nature-based ECE vs. traditional ECE on social and emotional outcomes.

Study Author and Year	Study Design	Sample Size (E/C)	Study Quality	Social Skills ⊕	Social & Emotional Development ⊕	Attachment ⊕	Initiative ⊕	Fewer Behavioural Problems ⊕
Cordiano et al. (2019) [26]	Controlled before & after	12/14	Weak	▼	-	-	-	▼ (favours control)
Müller et al. (2017) [47]	Controlled before & after	43/45	Weak	▲	-	-	-	▼ (favours control)
Agostini et al. (2018) [59]	Controlled before & after	41/52	Weak	-	▲	-	-	-
Cooper (2018) [56]	Controlled before & after	13/11	Weak	-	▼	▼	▼	-
Ernst et al. (2019) [29,30]	Uncontrolled before & after	78	Weak	-	-	▲	▲	-
Fyfe-Johnson et al. (2019) [33]	Controlled cross-sectional	20/13	Weak	▲	▲	-	-	▲
**Summary effect direction**				▲	▲	▲▼	▲▼	▼ (favours control)

Abbreviations: E = experimental; C = comparison; ECE= Early childhood education. GRADE—(assesses the certainty of evidence at an outcome level): **⊕** = Very low. **Effect direction:** Study level: ▲ = positive association with nature-based ECE; ▼ = negative association with nature-based ECE. Controlled before & after studies—difference between experimental and control group at follow-up (unless stated). Uncontrolled before & after studies—change since baseline (unless stated). Controlled cross sectional—difference between experimental and control (unless stated). Cross-sectional—positive, negative or no association. **Summary:** ▲ = studies show a positive association with nature-based ECE; ▼ = studies show a negative association with nature-based ECE; ▲▼ = conflicting findings. Summary effect direction considers study quality, design (i.e., controlled before and after weighted more than cross-sectional) and sample size.

**Table 3 ijerph-19-05967-t003:** Effect direction plot for nature-based ECE vs. traditional ECE on play.

Study Author and Year	Study Design	Sample Size (E/C)	Study Quality	Play Interaction ⊕	Play Disruption ⊕	Play Disconnection ⊕
Cordiano et al. (2019) [26]	Controlled before & after	12/14	Weak	▼	▼	▼
Burgess & Ernst (2020) [27,30]	Controlled before & after	84/24	Weak	▲	▲	▲
Robertson et al. (2020) [41]	Controlled cross-sectional	15/15	Weak	▲	-	-
**Summary effect direction**				▲	▲▼	▲▼

Abbreviations: E = experimental; C= comparison; ECE = Early childhood education. GRADE—(assesses the certainty of evidence at an outcome level): **⊕** = Very low. **Effect direction:** Study level: ▲ = positive association with nature-based ECE; ▼ = negative association with nature-based ECE. Controlled before & after studies—difference between experimental and control group at follow-up (unless stated). Uncontrolled before & after studies—change since baseline (unless stated). Controlled cross sectional—difference between experimental and control (unless stated). Cross-sectional—positive, negative or no association. **Summary:** ▲ = studies show a positive association with nature-based ECE; ▼ = studies show a negative association with nature-based ECE; ▲▼ = conflicting findings. Summary effect direction considers study quality, design (i.e., controlled before and after weighted more than cross-sectional) and sample size.

**Table 4 ijerph-19-05967-t004:** Effect direction plot for nature-based ECE vs. traditional ECE on cognitive outcomes.

Study Author and Year	Study Design	Sample Size (E/C)	Study Quality	Attention ⊕	Self-Regulation ⊕
Burgess & Ernst (2020) [27,30]	Controlled before & after	84/24	Weak	▲	-
Müller et al. (2017) [47]	Controlled before & after	43/45	Weak	▼	▲
Cooper (2018) [56]	Controlled before & after	13/11	Weak	-	▲
Ernst et al. (2019) [29,30]	Uncontrolled before & after	78	Weak	-	▲
Fyfe-Johnson et al. (2019) [33]	Controlled cross-sectional	20/13	Weak	▲	-
**Summary effect direction**				▲▼	▲

Abbreviations: E= experimental; C= comparison; ECE= Early childhood education. GRADE—(assesses the certainty of evidence at an outcome level): **⊕** = Very low. **Effect direction:** Study level: ▲ = positive association with nature-based ECE; ▼ = negative association with nature-based ECE. Controlled before & after studies—difference between experimental and control group at follow-up (unless stated). Uncontrolled before & after studies—change since baseline (unless stated). Controlled cross sectional—difference between experimental and control (unless stated). Cross-sectional—positive, negative or no association. **Summary:** ▲ = studies show a positive association with nature-based ECE; ▼ = studies show a negative association with nature-based ECE; ▲▼ = conflicting findings. Summary effect direction considers study quality, design (i.e., controlled before and after weighted more than cross-sectional) and sample size.

**Table 5 ijerph-19-05967-t005:** Effect direction plot for nature-based ECE vs. traditional ECE on nature connectedness.

Study Author and Year	Study Design	Sample Size (E/C)	Study Quality	Nature Relatedness/ Biophilia ⊕	Environmentally Responsible Behaviour ⊕	Awareness of Nature/ Environment ⊕
Elliot et al. (2014) [46]	Controlled before & after	21/22	Mod	▲	▼	-
Müller et al. (2017) [47]	Controlled before & after	43/45	Weak	▲	▼	-
Agostini et al. (2018) [59]	Controlled before & after	41/52	Weak	-	-	▲
Yilmaz et al. (2020) [65]	Uncontrolled before & after	40	Weak	▲	-	-
Barrable et al. (2020) [57]	Controlled cross-sectional	141/110	Weak	▲	▲	▲
Giusti et al. (2014) [54]	Controlled cross-sectional	11/16	Weak	▲	-	-
Rice & Torquati (2013) [34]	Controlled cross-sectional	68/46	Weak	■	-	-
**Summary effect direction**				▲	▲▼	▲

Abbreviations: E = experimental; C = comparison; ECE = Early childhood education. GRADE—(assesses the certainty of evidence at an outcome level): **⊕** = Very low. **Effect direction:** Study level: ▲ = positive association with nature-based ECE; ▼ = negative association with nature-based ECE; ■ = statistics not presented or no difference. Controlled before & after studies—difference between experimental and control group at follow-up (unless stated). Uncontrolled before & after studies—change since baseline (unless stated). Controlled cross sectional—difference between experimental and control (unless stated). Cross-sectional—positive, negative or no association. **Summary:** ▲ = studies show a positive association with nature-based ECE; ▼ = studies show a negative association with nature-based ECE; ▲▼ = conflicting findings. Summary effect direction considers study quality, design (i.e., controlled before and after weighted more than cross-sectional) and sample size.

## Data Availability

Not applicable.

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
