# Peer review of "Nature-Based Early Childhood Education and Children’s Social, Emotional and Cognitive Development: A Mixed-Methods Systematic Review"

_ijerph, 2022, doi:10.3390/ijerph19105967_

Round 1

Reviewer 1 Report

Thank you to the authors for their work on this paper. It is focused on an area of great importance and makes a clear case for the significance of this research, as well as what further research is needed. Overall the paper provides comprehensive and clear insights about the analysis undertaken and the findings.

Some expansion on the reference to source [2] in lines 42-44 may be helpful; at present this is somewhat vague. 

The aims as stated in lines 90 - 94 may benefit from some minor editing to ensure clarity - was it an improvement to development that was being looked at, or a positive affiliation? This is evident from further information provided elsewhere in the paper but would benefit from clarifying at this point. 

One area which may benefit from some further exploration relates to the study quality ratings and the prevalence of "weak" ratings - the point regarding why this rating was allocated is generally clear; it may be worth clarifying whether work was done to assess how the methodological limitations were addressed by the respective authors and whether this provided any mitigation around quality. 

Research ethics should also be addressed; ideally in 2. Materials and methods (lines 100 - 109 currently). 

Thanks again to the authors for their thorough preparation of this paper and for undertaking this research. I appreciated the opportunity to review and read through their analysis and insights. 

Author Response

Dear reviewer, thank you for taking the time to review our manuscript and providing useful feedback which will further enhance our review. We have provided a response to your review below.

  1. Thank you for your comment. This sentence now reads, “However, according to a global report by the World Health Organization (WHO), evidence suggests that cognitive, social and emotional outcomes are often low in children, and typically worse in those with lower socio-economic status (SES) [2].” We hope this reference is now clearer. (line 43).
  2. Thanks for your comment. Our research questions present a two-tailed research aim that is open to null or negative associations (i.e. we do not project a direction of association) rather than searching for improvements/positive associations. Therefore, we have opted to keep our original questions. We hope the reviewer understands our decision.
  3. Thank you for your comment. We do acknowledge that authors of eligible studies are likely to both recognise and mitigate risks in their studies. The purpose of the EPHPP tool – a robust, reliable and valid assessment of study quality at an individual level – is to provide a standardised rating on the study quality based on the criteria set by EPHPP to aid readers in interpretation of the eligible studies. The EPHPP tool does consider how primary study authors mitigate limitations which then is reflected in the study quality rating. For example, in cross-sectional studies the extent to which relevant confounding variables have been considered. We have endeavoured to be transparent about study quality by a) giving the domain and overall rating of each eligible study (supplementary material) and b) providing the quality rating across the studies (Figures 2 – 4). We hope that this provides sufficient detail to the readers and enables researchers to understand where improvements can be made in future studies.  In addition, it also important to look beyond the individual study quality to the quality of the evidence, which we have provided through GRADE in this review. Therefore, we feel the addition of further information on study quality is beyond the scope of this systematic review and would dilute the key message of raising the quality of studies by, for example, moving away from cross-sectional designs. However, we do agree that tools to assess quality in this field are not perfect and have therefore added the following, “As noted in a conceptually similar systematic review looking at immersive nature experiences on children’s health, it is important to highlight that in this field these tools (EPHPP and others) which assess study quality may provide weak ratings in certain domains, e.g. blinding, where it is difficult to conduct research practices that is the gold standard [11]” (line 1086 ). This is then followed by a critique of GRADE in the next section. We hope that you are satisfied with this change.
  4. This is a systematic review involving synthesis of primary research studies and, therefore, does not require ethical approval. This is clarified in line 1193 – “Institutional Review Board Statement: Not applicable.”

Reviewer 2 Report

The systematic review presented in the research work is very interesting. The article is properly written and well structured. Findings confirm that further research should be carried out.

Comments and suggestions

Minor Format issues

  • Lines 19 to 25.- Lines are not properly aligned.
  • Line 21.- Studies were eligible if a) children … A colon is required before i)
  • Line 122.- including i) children …  A colon is required before i)
  • Line 157.-  4. Child … An extra space should be removed
  • Line 262.- These were i)… A colon is required before i)
  • Lin 328.-  studies [55] , … An extra space should be removed.
  • Line 423.- r (p= 0.11, η2= 0.03) [44]. The other study …  [44]
  • Line 532.-  56] ; n=1 ECE natural … An extra space should be removed
  • Line 622.-  [54]. Finally, …… An extra space should be removed
  • From line 632 to line 744.-  justification is different.
  • Line 822.-  Figure 7 … The arrows are not properly displayed
  • Line 854.-  Figure 8 … The arrows are not properly displayed
  • Line 855.- the text should be located above Figure 8
  • Line 884.- Figure 9 … The arrows are not properly displayed
  • Line 906.- Figure 10 … The arrows are not properly displayed
  • Line 1101.- Figure emotional, and … An extra space should be removed
  • Line 1130.- Early childhood education … All the three words in capital letters? … Sometimes in the text is written in capital letters (i.e., line 19) and in other places in lower letters (i.e., 1130).
  • Lines 1129 to 1139.-  The list of abbreviations can be deleted, since the acronyms are previously defined in the paper. Line 1134 à  sign) … The final bracket should be removed

Some items to clarify in the paper

  • Abstract (Line 24 to line 28), compared with the Data included in Figure 1.-  Abstract writing could be reworded and reordered to make it easier and making it consistent with results shown in Figure 1.
  • Figure 1 is a little bit confusing… Numbers should be clarified
    • Social and emotional development Social and emotional     n=13
    • Social and emotional development Play                                   n=10
    • Cognitive development n=11
    • Nature connectedness                                 n=9
    • Qualitative n=10   

             a + b + c + d = 43

              a + b + c + d + e = 53;    Data in the previous box 59 (49 +10)   ….?

  • Figure 6 (line 783).- No findings are showed in the figure about ‘Theme 4’ (Restorative and invigorating effect of nature), despite they are included in the text (lines 567 … importance of the natural playground in helping children invigorate and/or restore their energy for the diverse types of play children engage in).
  • The abstract can be reworded to match in a better way the conclusions (section 5).

Bibliography

  • Doi should be added to all the references.

Author Response

We would like to thank you for reviewing and providing useful and clear feedback. We have provided a point by point response to your comments below.

Minor Format issues
•    Lines 19-25 are now aligned.
•    Line 21. Colon now added.
•    Line 122. Colon now added.
•    Line 157. Space removed.
•    Line 262. Colon added. 
•    Line 328. Space removed. 
•    Line 423. Citation updated. 
•    Line 532. Space removed. 
•    Line 622. Space removed. 
•    Line 632 – Line 744. Justification updated. 
•    Line 822 – Figure 7 now updated.   
•    Line 854 – Figure 8 now updated.   
•    Line 855. Text now located above.
•    Line 884 – Figure 9 now updated.   
•    Line 906 – Figure 10 now updated.   
•    Line 1101. Space removed. 
•    Line 1130. Updated to capital letters 
•    Lines 1129 to 1139. The Editor can decide whether the list of abbreviations should be removed. 

Some items to clarify in the paper

  1. See the comment below, we hope that eligible studies per outcome is clearer now. 
  2. Thanks for your comment. The authors agree that the figure could be clearer. This systematic review is part of a larger review that also includes physical outcomes, hence the larger value here. This is explained in the text, “A total of 59 unique studies (representing 65 individual papers) met the inclusion criteria, of which, 36 studies (representing 40 individual papers) included a social and emotional, cognitive, and/or nature connectedness outcome (26 quantitative; 9 qualitative; 1 mixed-methods).” We appreciate that the Figure should be understandable in isolation so we have made a small tweak to ensure this is more clear. This now reads “A total of 59 unique studies (representing 65 individual papers) met the inclusion criteria (including the physical outcome domain which is presented in another paper)...” We hope this is clearer now (line 310).    
  3. Thanks for your comment. Figure 6 presents the findings from the qualitative synthesis in which higher-order themes are shown on the left-hand side and lower order themes are shown on the right. For Theme 4, restorative and invigorating effect of nature, there are no lower-order themes, just a higher-order one. The text on line 756 provides more detail of this higher-order theme. We hope that this is now clearer. 
  4. Thanks for your comment. The conclusion in the abstract reads, “Nature-based ECE may improve some childhood development outcomes, however, high-quality experimental designs describing dose and quality of nature are needed to explore the hypothesised pathways connecting nature-based ECE to childhood development.”  Section 5 in the manuscript text provides a more comprehensive conclusion, “To test this hypothesis and understand more about the mechanisms in which nature-based ECE impacts children’s social, emotional and cognitive development, the evidence base must move to higher-quality study designs that adequately describe the nature exposure more precisely and with robust methods and over a longer duration. This will elevate the current evidence-base and inform research, policy and practice on the complex pathways in which nature-based ECE impacts children’s social, emotional, and cognitive development.” We acknowledge the omission of details about the association between nature-based ECE and outcomes and reference to policy, however, given the limitations with word count (200 words) we could not add this level of detail in the abstract. 

Bibliography

We agree that DOI’s should be added to references, but we will leave this to the journal. 

Reviewer 3 Report

In recent years systematic review and meta analysis is being used in an increasingly number of research fields. In many of these fields these studies are not often done in a high quality manner. I was so impressed by detailed work that the authors have put into this paper that I will be recommending one of my students working in the field of ECE to read this as soon as it is published.  The authors should be quite proud of the work that has gone into the project and how easily accessible and easily understood that they have made the writing of the paper. I also appreciate that this is a mixed review that includes both qualitative and quantitative studies. While it seemed the extracted data could not allow the authors to conduct a meta analysis, I do appreciate the use of the effect direction plots (tables). I only have some very minor suggestions for the authors regarding adding one citation to support a claim, fixing some citations, and also reinserting clearer figures. Other than these, I really do feel this paper is ready for publication. Congratulations on this fine work.  Please find the attached pdf. 

Author Response

Dear reviewer, 

We would like to thank you for taking the time to review our systematic review and for providing positive and supportive feedback. As advised, we have made the minor tweaks to adding/ amending citations, updated figures and removing spacing. 

Thank you again. 

Reviewer 4 Report

Thanks for inviting me to review this interesting review article. The topic is really attractive and worthwhile. The review was well conducted, and the article was well written. I really enjoy reading it. However, I have some suggestions for the authors to improve the readability and rigor of this research paper.

First, please state a guiding research question for this review, which could be addressed by the quantitative and qualitative parts of this study.

Second, the literature review part could be improved to better justify this study. For example, there are arguments about the effectiveness of nature-based curriculum. This study can help settle the debate.

Third, please provide the reference about the PI(E)COS (Population, Intervention or Exposure, 111 Comparison, Outcomes and Study design) framework. 

Fourth, please follow the APA publication manual, especially those about "Journal Article Reporting Standards for Qualitative and Review Studies'. Please refer to Table 2 "Qualitative Meta-Analysis Article Reporting Standards (QMARS): Information Recommended for Inclusion in Manuscripts That Report Qualitative Meta-Analyses" for details. The effect size of all the reviewed quantitative studies should be presented in a table.

Last, there are so many figures. I am wondering whether you can combine them or provide the path coefficients for the figures. 

Author Response

Dear reviewer, we thank you for taking the time to review our manuscript and providing useful feedback that will further improve the study. We have provided a point-by-point response to your comments. 

  1. We have two guiding research questions. The first research question speaks to the quantitative part of the study (e.g. use of association), and the second research question speaks to the qualitative part of the study (e.g. exploring perceptions).
  2. Thanks for your comment. We have added some text to strengthen our argument, as you suggest. Lines 84-88 now reads “By conducting a systematic review of nature-based ECE on children’s health and development specifically, this will inform future research needs, synthesise and summarise the global evidence, inform national and international policy, and enable the field to consider and explore the potential mechanisms by which certain child-health outcomes might improve”. 
    Further on, line 93-95 now reads “By synthesising both quantitative and qualitative evidence, this review can provide a comprehensive understanding of the evidence base that could be pivotal to informing future research, policy and practice in the nature-based ECE field”. We hope that this “sells” the importance of our systematic review more. 
  3. We have now inserted a citation for the PICOS framework (line 122).  
  4. Thanks for your comment and for drawing our attention to “Journal Article Reporting Standards for Qualitative and Review Studies”. We agree this is a highly important source to elevate the reporting of both qualitative and mixed methods systematic reviews. As noted in Lines 117-118, we have followed the adapted PRISMA for reporting systematic reviews of qualitative and quantitative evidence. This is an equally important resource for ensuring quality when reporting on mixed-methods systematic reviews. Furthermore, PRISMA is widely used in our field and recommended by the Cochrane Collaboration who produce the highest quality systematic reviews. Finally, all study effect sizes are included in supplementary file 5. Given this is a large systematic review with multiple outcome domains, we felt that effect sizes per study could clutter the tables and would be better placed here.
  5. We share the reviewer's concern of the number of Figures in the manuscript. Figure 7-10 specifically present the potential factors that might explain why nature-based ECE could improve certain outcomes. These are hypothesised pathways only, where we have used the qualitative themes to explain why quantitative outcomes have occurred. As these are hypothesised pathways only, we, therefore, cannot use statistical analysis to provide the path coefficients unlike, for example, in a single longitudinal study. We debated putting these Figures in the supplementary file, but we feel a graphical representation aids the reader in understanding this novel mixed-methods systematic review. If the Editor feels that there are too many Figures in this manuscript we would also be happy to move these to the Supplementary Material. We hope the reviewer understands our decision here.